# A Review of Biomarkers and Their Clinical Impact in Resected Early-Stage Non-Small-Cell Lung Cancer

**DOI:** 10.3390/cancers15184561

**Published:** 2023-09-14

**Authors:** Weibo Cao, Quanying Tang, Jingtong Zeng, Xin Jin, Lingling Zu, Song Xu

**Affiliations:** 1Department of Lung Cancer Surgery, Tianjin Medical University General Hospital, Tianjin 300052, China; caoweibo@tmu.edu.cn (W.C.); tangquanying@tmu.edu.cn (Q.T.); xiaozeng007@tmu.edu.cn (J.Z.); jinxin@tmu.edu.cn (X.J.); zulingling@tmu.edu.cn (L.Z.); 2Tianjin Key Laboratory of Lung Cancer Metastasis and Tumor Microenvironment, Lung Cancer Institute, Tianjin Medical University General Hospital, Tianjin 300052, China

**Keywords:** non-small-cell lung cancer, early-stage, surgery, biomarkers, prognosis

## Abstract

**Simple Summary:**

Non-small-cell lung cancer (NSCLC) accounts for approximately 85% of all lung cancers, and the postoperative survival of early-stage NSCLC patients remains unsatisfactory. Over the last several decades, mutant genes, immunological checkpoints, and blood-based biomarkers have been developed and tested to have diverse effects on the survival of early-stage NSCLC. Herein, we reviewed the pertinent literature to determine the prognostic effect of related indicators on early-stage NSCLC, and we will accurately predict patient outcomes and guide patient treatment in the future.

**Abstract:**

The postoperative survival of early-stage non-small-cell lung cancer (NSCLC) patients remains unsatisfactory. In this review, we examined the relevant literature to ascertain the prognostic effect of related indicators on early-stage NSCLC. The prognostic effects of the epidermal growth factor receptor (EGFR), anaplastic lymphoma kinase (ALK), mesenchymal–epithelial transition (MET), C-ros oncogene 1 (ROS1), or tumour protein p53 (TP53) alterations in resected NSCLC remains debatable. Kirsten rat sarcoma viral oncogene homologue (KRAS) alterations indicate unfavourable outcomes in early-stage NSCLC. Meanwhile, adjuvant or neoadjuvant EGFR-targeted agents can substantially improve prognosis in early-stage NSCLC with EGFR alterations. Based on the summary of current studies, resected NSCLC patients with overexpression of programmed death-ligand 1 (PD-L1) had worsening survival. Conversely, PD-L1 or PD-1 inhibitors can substantially improve patient survival. Considering blood biomarkers, perioperative peripheral venous circulating tumour cells (CTCs) and pulmonary venous CTCs predicted unfavourable prognoses and led to distant metastases. Similarly, patients with detectable perioperative circulating tumour DNA (ctDNA) also had reduced survival. Moreover, patients with perioperatively elevated carcinoembryonic antigen (CEA) in the circulation predicted significantly worse survival outcomes. In the future, we will incorporate mutated genes, immune checkpoints, and blood-based biomarkers by applying artificial intelligence (AI) to construct prognostic models that predict patient survival accurately and guide individualised treatment.

## 1. Introduction

Lung cancer mortality is predominant among cancer patients worldwide [1], and approximately 85% of lung cancers are non-small-cell lung cancer (NSCLC) [2]. The comprehensive treatment paradigm for early-stage NSCLC has changed rapidly over the past 20 years, and radical resection of cancerous lesions remains the preferred treatment option for early-stage NSCLC [3].

Nevertheless, postoperative survival of patients remains unsatisfactory, with five-year survival rates for stage I to III lung cancer ranging from 90% to 12% [4]. With the improvement in molecular testing, more mutant genes have been detected in NSCLC, such as the epidermal growth factor receptor (EGFR) and Kirsten rat sarcoma viral oncogene homologue (KRAS) genes [5,6]. Immune checkpoint inhibitors (ICIs) target multiple immune checkpoints, including the programmed death ligand 1 (PD-L1) and the programmed cell death protein-1 (PD-1) [7,8]. Patients with lung cancer recurrences often indicate tumour micro-metastases indexes in the blood circulation, which can often reflect the molecular characteristics of micro-metastases.

In this review, we included relevant literature to discuss further the prognostic value and clinical significance of related biomarkers in early-stage NSCLC patients.

## 2. Methods

The literature from PubMed and published in English was reviewed. We conducted the literature search using a combination of the following keywords: “Carcinoma, Non-Small-Cell Lung” OR “Adenocarcinoma of Lung” OR “Squamous cell lung cancer” AND “Surgical Procedures, Operative” AND “Mutation” OR “Programmed cell death-1” OR “Programmed cell death-ligand 1” OR “Tumor mutational burden” OR “Neoplastic Cells, Circulating” OR “Biomarkers”. Detailed retrieval strategies are listed in Appendix A. The deadline for the literature search was 1 February 2023. The included literature was selected from studies related to the theme of the review.

## 3. Results

### 3.1. Genetic Alterations

With an improvement in molecular detection techniques, increasing genetic alterations have been detected in NSCLC. Therefore, we summarised the effects of genetic modifications on the prognoses of resectable early-stage NSCLC (Table 1).

#### 3.1.1. EGFR Alterations

The EGFR belongs to the tyrosine kinase receptors (TKRs) superfamily, which also includes ErbB2/human epidermal growth factor receptor 2 (HER2), ErbB3/HER3, and ErbB4/HER4 [52]. EGFR binds to its ligand and activates downstream signalling pathways involved in cell proliferation, migration, and survival. It has been shown that EGFR is critical for the biology of epithelial-derived malignancies, and therefore, potential novel inhibitors targeting EGFR deserve extensive exploration [53,54].

EGFR genetic alteration rates are standard in all pathological types of NSCLC, particularly in LUAD [55,56]. Meanwhile, many EGFR alteration subtypes and molecular changes have been found in early-stage NSCLC. Regarding stage II–III nonsquamous NSCLC patients receiving surgery and platinum-based adjuvant chemotherapy, it was found that EGFR alteration was an independent predictor for shorter relapse-free survival (RFS) and tumour recurrence [9]. For stage I–III resected LUAD, the risk of brain and bone metastases was higher in patients with EGFR alterations. However, in all patients included, the RFS of patients with EGFR alterations was not different from those with wild-type EGFR. Alterations in EGFR only had a statistically shorter RFS than the wild-type group in patients with solid nodules, stage II–III, or acinar/papillary/invasive mucinous predominantly LUAD [10]. Even in patients with pN0-1M0 LUAD, favourable EGFR alterations were indicated as risk factors for postoperative recurrence. The recurrence risk stratification was dependent upon the pathological stage and degree of histological malignancy [11]. When assessing the molecular changes in EGFR, its amplification was associated with poorer RFS in early-stage EGFR-mutant LUAD [12], and a higher level of EGFR amplification correlated with poorer survival in surgically treated NSCLC patients [13].

However, Isaka et al. found that EGFR alterations were significantly associated with favourable RFS in patients with recurrent LUAD after surgical resection, and a longer median RFS was observed in patients with EGFR alterations (20.2 months) compared to wild-type patients (13.3 months) [14]. It has been confirmed that patients with mutant EGFRs had a greater chance of survival than those with wild-type EGFRs for resectable NSCLC [15,16]. Additionally, the prognostic effect of alterations in EGFR for early-stage NSCLC differed even for the pathological stage. EGFR alterations did not impact the survival of stage IA NSCLC but were favourable prognostic factors for better disease-specific survival (DSS) and OS in patients with stage IB NSCLC [17]. Based on the original studies above, EGFR alterations’ prognosis in early-stage resected NSCLC remains controversial.

When alterations in the EGFR subtypes were studied, it was found that EGFR alterations in exons 18–19 (five-year survival: 100%) represented better survival outcomes in resected NSCLC than those without such alterations (five-year survival: 47%) [18]. The prognostic effect of exon 21 L858R alterations and EGFR exon 19 deletions on resected NSCLC patients differed by stage. Exons 19 and 21 did not significantly differ in median OS for location I patients. However, exon 19 deletions had an exceptionally favourable OS compared to exon 21 L858R alterations in stage II and III patients [19]. Furthermore, for pN1-N2 LUAD, patients with exon 19 deletions had longer disease-free survival (DFS) and OS than those with exon 21 L858R alterations [20]. However, a study of resectable LUAD showed that exon 19 deletions were more likely to have an extrathoracic recurrence and significantly shortened RFS when compared to exon 21 L858R alterations, suggesting that exon 19 deletions were poor prognostic factors [21]. EGFR T790 alterations tended to be independent of shorter RFS and OS in resected NSCLC [22]. Phosphorylated EGFR (pEGFR) could independently predict shorter survival in stage I NSCLC patients undergoing surgery [23].

Since EGFR alterations are common in NSCLC patients, targeted agents for EGFR alterations have also been extensively studied. The ADAURA trial found that patients with stage IB–IIIA, EGFR-mutated NSCLC (*n* = 682) who received postoperative adjuvant osimertinib had better DFS compared to placebo (four-year DFS: osimertinib 73% vs. placebo 38%) [57]. The updated data from the ADAURA trial revealed that patients in the osimertinib group had more prolonged OS than the placebo group (five-year OS: osimertinib 88% vs. placebo 78%) [58]. Hence, adjuvant osimertinib provided a survival benefit for IB–IIIA NSCLC with EGFR alterations. Another clinical trial (ADJUVANT) reported that for NSCLC with stage II–IIIA EGFR alterations, adjuvant gefitinib (28.7 months) resulted in significantly longer median DFS than vinorelbine plus cisplatin (18 months) [59]. In the study (Emerging-CTONG 1103) involving IIIA (N2) resectable NSCLC patients with EGFR alterations, investigators examined the efficacy of neoadjuvant/adjuvant erlotinib and gemcitabine plus cisplatin (GC chemotherapy). Although neoadjuvant erlotinib (54.1%) had a higher objective response rate (ORR) than GC chemotherapy (54.1%), the difference was not statistically significant (*p* = 0.092). The median progression-free survival (PFS) with erlotinib (21.5 months) was substantially longer than that with GC chemotherapy (11.4 months) [60]. Therefore, in early-stage NSCLC patients with EGFR alterations, adjuvant or neoadjuvant EGFR-TKIs can significantly prolong patient survival compared to placebo or platinum-based chemotherapy.

#### 3.1.2. KRAS Alterations

KRAS, belonging to the RAS gene family, is activated by binding to GTP and triggers many cellular activation processes, including transcription, translation, cell survival, and apoptosis [61].

KRAS G12C (KRAS^G12C^) was significantly associated with a worsening DFS in stage I–III resectable LUAD. However, patients with the KRAS-mutant (KRAS^MUT^) and the wild-type KRAS (KRAS^WT^) did not differ in DFS [24]. Another study also confirmed that NSCLC patients (Stage I–III) with KRAS^MUT^ had similar OS to those with KRAS^WT^ [25]. KRAS^MUT^, as an independent prognostic factor, was correlated with inferior DFS and OS compared to KRAS^WT^ in stage I LUAD [26]. Meanwhile, KRAS alterations were associated with worsening DFS, OS, and a higher recurrence risk in early-stage LUAD, especially in tumours with predominantly solid components [27,28]. Regarding early-stage resected NSCLC, KRAS alterations were prognostic determinants for worsening survival [16]. Conversely, Ayyoub et al. showed that KRAS alterations were associated with superior DSS in resected NSCLC [29].

Subgroup analyses of a meta-analysis involving NSCLC patients (*n* = 6939) revealed that KRAS alterations predicted an unfavourable OS for stage I–IIIA NSCLC [62]. In general, KRAS genetic alterations predicted adverse outcomes in resected NSCLC.

#### 3.1.3. ALK Alterations

Anaplastic lymphoma kinase (ALK), a transmembrane receptor tyrosine kinase (RTK), has a highly homologous sequence to insulin receptor kinases. Oligomerase is the primary driver of ALK’s downstream signalling, and fusion proteins such as echinoderm microtubule-associated protein-like 4 (EML4)-ALK and nucleophosmin (NPM)-ALK are critical components of the downstream signalling pathway’s activation [63,64].

For patients with stage I LUAD undergoing surgery, ALK rearrangement exhibited a significantly worsening RFS but did not affect OS [30]. Furthermore, patients with ALK alterations experienced worse RFS than those with EGFR alterations in early-stage resected NSCLC [31]. However, a study published by Matsuura et al. confirmed that ALK rearrangement was a more favourable biomarker for cancer-specific survival (CSS) and OS than non-ALK rearrangement in resected NSCLC [32]. This may be attributed to the effect of ALK-targeted inhibitors. A different study demonstrated that EML4-ALK mutant 3 had a substantially shorter DFS in patients with resected NSCLC [33]. However, several studies concluded that ALK rearrangement had no predictive value in surgically resected NSCLC [34,35]. In summary, there were no conclusive conclusions about the prognostic effect of ALK rearrangement on early-stage NSCLC.

#### 3.1.4. MET Alterations

The mesenchymal–epithelial transition (MET) gene is an oncogene that encodes RTK, which binds to the hepatocyte growth factor (HGF) and promotes the aggressive nature of the tumours by inducing angiogenesis [65]. C-MET had diverse prognostic outcomes on resectable LUAD (Stage IB–IIIA) depending on EGFR alteration status. C-MET significantly correlated with reduced RFS and OS in EGFR-negative groups but not EGFR-positive groups [36]. Resectable NSCLC patients with MET-increased gene copy number had a higher risk of death and shorter survival [37]. MET amplification and overexpression have been found in NSCLC, but they did not affect survival in stage I–III NSCLC [38]. For patients with resected stage I–IIIA LUAD, there was no difference in survival between MET exon 14 (METex14) skipping and the other significant genetic alterations [39].

#### 3.1.5. ROS1 Alterations

The C-ros oncogene 1 (ROS1) encodes RTK, but its corresponding ligand is unrecognised, and therefore, our understanding of the function of ROS1 is limited [66]. Kim et al. indicated that the ROS1 fusion gene was an independent biomarker for disease recurrence in resectable stage I–II LUAD [40]. However, Chen et al. indicated no difference in survival (*p* = 0.555) between the ROS1 fusion-positivity and fusion-negativity groups in early-stage LUAD, but this might be attributed to the limited number of ROS1 fusion individuals [41].

#### 3.1.6. TP53 Alterations

Tumour protein p53 (TP53) is a tumour antioncogene, which encodes the p53 tumour suppressor protein and functions to protect cellular DNA, as well as affecting cell metabolism, differentiation, and apoptosis [67]. The TP53 alteration was an independent predictor for poor RFS in resected EGFR-mutant LUAD [12]. Compared with a single EGFR alteration, a single TP53 alteration as a negative biomarker had a shorter RFS in stage I–II LUAD [40]. Jao et al. concluded that TP53 also leads to worse DFS and OS in resectable NSCLC [42]. A different study confirmed that TP53 did not correlate with DFS in stage I LUAD and LUSC but only conferred reduced OS in stage I LUAD, not in LUSC [43].

The study by Zhou et al. further demonstrated that p53 pathway alterations correlated with poor DFS in surgically treated LUAD [44]. Regarding stage I–III NSCLC patients, p53 alterations (Exons 5-8) had an adverse prognostic effect on survival [45,46]. Hence, the prognosis of TP53 alterations in resected NSCLC remains debatable.

#### 3.1.7. Other Genetic Alterations

Rearranged during transfection (RET) as an oncogene expresses cellular surface receptors, and its ligand is a neurotrophic factor derived from glial cell lines. The RET gene is involved in kidney and nervous system development and even the formation of tumours [68]. RET fusion genes, as prognostic biomarkers, were correlated with tumour recurrence in early-stage resected LUAD [40]. The V-RAF murine sarcoma viral oncogene homolog B1 (BRAF) gene encodes the BRAF protein. It is involved in the mitogen-activated protein kinase (MAPK) pathway, which regulates cell proliferation, differentiation, and apoptosis [69]. BRAF alterations were independent biomarkers for improved DFS and OS in resectable LUAD [28].

For patients with resected LUSC, a significant association was found between PIK3CA alterations and longer OS and a longer time to recurrence (TTR) [47]. However, Song et al. concluded that PIK3CA alterations were correlated with worsening OS in stage I–IIIA resected LUAD [48]. Another study also demonstrated that PIK3CA correlated with poor survival in stage I NSCLC [49]. Furthermore, thyroid transcription factor-1 (TTF-1) gene amplification was a prognostic predictor for worsening DFS and OS in resected LUAD [50]. When considering resected EGFR-mutant LUAD, patients with protein kinase D (PKD)/TP53 co-alterations had shorter DFS than those without co-alterations [51].

### 3.2. Immune-Related Indicators

Tumour-induced immunosuppression encourages tumours to escape immune surveillance, and immunotherapy can activate the immune system and induce a sustained anti-tumour immune response. Several essential immune checkpoints, including PD-L1, PD-1, programmed death-ligand 2 (PD-L2), and T-cell immunoglobulin and mucin-domain-containing-3 (TIM-3), were found to be targets of ICIs. Herein, we included immune-related indicators to explore their prognostic relevance in resected NSCLC (Table 2).

#### 3.2.1. PD-L1

Patients with PD-L1 overexpression had significantly inferior survival in stage I–III resectable LUAD, especially in acinar/papillary-dominant LUAD [70,71,72]. Takada et al. indicated that the 1% cut-off level of PD-L1 was more sensitive in predicting postoperative prognosis than the 5% level. Early-stage LUAD patients with positive PD-L1 were correlated with poor DFS and OS at the 1% cut-off level [73]. For patients with stage I–III LUSC (*n* = 205), PD-L1 was also an effective marker for worsening OS at the 1% cut-off level [74]. A study by Azuma et al. examined the relationship between PD-L1 and EGFR alterations in stage I–IIIB NSCLC. The high expression level of PD-L1 correlated with EGFR alterations caused by EGFR signalling. Furthermore, high PD-L1 expression was a harmful indicator for OS (Median OS: high 55.9 months vs. low 72.6 months) [75]. Similarly, PD-L1 ≥ 1% was correlated with worsening DFS and OS in stage I–IIIA EGFR-mutated NSCLC [76]. Regarding the prognostic effects of CD8 (cut-off values: 543 per mm^2^) and PD-L1 (cut-off values: 1%) in resected NSCLC, CD8 positivity/PD-L1 negativity was favourable for RFS and OS [77]. Handa et al. also verified that NSCLC patients (Stage I) with high levels of PD-L1 (>5%) had a worse prognosis than patients with low levels of PD-L1 (five-year RFS: high 63.4% vs. low 81%, *p* = 0.061). When considering surrounding immune cells, the intratumoural CD8 level influenced the predictive value of PD-L1. It concluded that PD-L1 had no predictive effect on survival in high-level cohorts of intratumoural CD8, but overexpression of PD-L1 predicted significantly poor RFS in cohorts with low-level CD8 (five-year RFS: PD-L1 (>5%) 63.4% vs. PD-L1 (≤5%) 81%, *p* = 0.034) [78].

Stage I LUAD patients with high PD-L1 expression (≥5%) had a favourable prognosis for RFS but not OS [79]. Regarding stage IIB–IIIB LUAD, Gross et al. concluded that patients with high PD-L1 expression (≥1%) on tumour cells and tumour-associated macrophages (TAMs) received survival benefits from adjuvant chemotherapy [89]. Similarly, PD-L1 was correlated with a longer OS in early-stage NSCLC [80]. Teramoto et al. indicated that the prognostic effect of PD-L1 differed by the pathological stage. Postoperative RFS in location I NSCLC with high PD-L1 expression was significantly prolonged over those with low PD-L1 expression (five-year RFS: high 94.1% vs. low 75.1%, *p* = 0.031). Conversely, for patients with stage II–IIIA NSCLC, high PD-L1 expression was prone to postoperative recurrence [81]. However, Song et al. validated that PD-L1 was not correlated with DFS and OS in resected LUAD [82].

A meta-analysis involving 15 studies predicted that PD-L1 as an independent prognostic indicator was related to unfavourable DFS and OS [83]. In conclusion, high levels of PD-L1 indicated poor survival in resected NSCLC compared to low levels.

#### 3.2.2. Other Immune-Related Markers

For early-stage LUAD patients, overexpression of PD-L2 (>1%) significantly predicted worsening DFS and OS [72]. TIM-3 positivity on tumour cells (≥24%) and tumour-infiltrating lymphocytes (TILs, ≥11%) was predictive of poor RFS and OS in early-stage LUAD. Subgroup analyses were conducted to investigate TIM-3/PD-1/CD8 expression levels for the prognosis of LUAD. The TIM-3 positivity/PD-1 positivity/CD8 negativity subgroup had the worst prognosis, whereas the TIM-3 negativity/PD-1 negativity/CD8 positivity subgroup had the best forecast [84]. Regarding early-stage nonsmoking patients, tumour mutation burden (TMB) was a poor prognostic indicator for OS [85]. Conversely, another study demonstrated that TMB had no prognostic value on stage I–II resected LUAD [86]. High OX40 (CD134) expression on TILs tended to unfavourable RFS and OS in location I NSCLC [87]. However, OX40, expressed on the surface of tumour-immune infiltrated cell membranes, correlated with favourable OS in stage I–III NSCLC [88].

We have summarised the prognostic significance of immune checkpoints for NSCLC patients. Meanwhile, we included relevant clinical trials to investigate the effect of ICIs on the prognosis of early-stage NSCLC. An RCT (IMpover010) enrolled 1280 NSCLC patients undergoing radical resection (IB–IIIA). After receiving four cycles of platinum-based chemotherapy, patients were randomised (1:1) to receive either 16 processes of adjuvant atezolizumab (PD-L1 inhibitor) or optimal supportive care. Finally, atezolizumab significantly prolonged DFS in stage II–IIIA patients with PD-L1 on tumours ≥ 1% (HR: 0.66; 95% CI: 0.50–0.88; *p* = 0.0039) and in all patients with stage II–IIIA (HR: 0.79; 95% CI: 0.64–0.96; *p* = 0.020) compared to optimal supportive treatment [90]. Another RCT (Checkmate 816) included resectable stage IB–IIIA NSCLC patients who underwent neoadjuvant nivolumab (PD-1 inhibitor) plus platinum-based chemotherapy (*n* = 179) or platinum-based chemotherapy alone (*n* = 179). The median event-free survival (EFS) was considerably longer in the neoadjuvant nivolumab plus chemotherapy group (31.6 months) than in the neoadjuvant chemotherapy group (20.8 months). The pathological complete response (pCR) rates were 24% and 2.2% in both groups, respectively [91]. A single-arm trial included 30 resectable NSCLC patients (Stage IB–IIIA) receiving neoadjuvant atezolizumab, carboplatin, and paclitaxel. A high proportion of patients (17/30, 57%) had a primary pathologic response (MPR) [92]. Although PD-L is a poor prognostic factor, adjuvant PD-L1 inhibitors can prolong survival in early-stage NSCLC. Additionally, the prophetic role of PD-1 is currently unknown, but neoadjuvant inhibitors of PD-1 can extend patient EFS.

### 3.3. Blood-Based Biomarkers

Although radical resection remains the preferred treatment option for early-stage NSCLC, the recurrence rates for lung cancer are still very high, indicating that micro-metastases still exist in the perioperative period. Therefore, we included studies on blood biomarkers, such as circulating tumour cells (CTCs), routine blood examinations, circulating nucleic acids, and tumour markers, to explore their prognostic value on early-stage NSCLC (Table 3).

#### 3.3.1. CTCs

CTCs shed from the primary cancerous lesion and circulating in the bloodstream can result in tumour recurrence or distant metastases [162]. Several prospective studies have shown that preoperative peripheral venous CTCs were associated with unfavourable DFS and OS in resectable NSCLC [93,94]. Concerning peripheral folate receptor-positive CTCs (FR^+^ CTCs) in resected NSCLC, their high preoperative levels correlated with poor RFS and OS [95]. Meanwhile, in a study including patients with stage I–IIIA NSCLC (*n* = 30), a value of ≥3 for PD-L1^+^/EMT^+^ CTCs in preoperative peripheral blood predicted poor RFS and OS, but RFS failed to reach statistical differences, most likely attributed to the small sample size [96]. Miguel-Pérez et al. compared the prognostic effect of peripheral CTCs in LUAD and LUSC and concluded that postoperative CTCs were only associated with poor RFS and OS in LUAD [97]. Additionally, postoperative peripheral thyroid transcription factor-1 CTCs (TTF-1^+^ CTCs) predicted a worse PFS on early-stage NSCLC [98]. Li et al. revealed that preoperative and postoperative LUNX^+^ CTCS in peripheral blood were independent predictors of unfavourable DFS and OS in resected NSCLC [99]. Moreover, Crosbie et al. collected blood samples (10 mL) from the tumour-draining pulmonary veins of stage I–IIIA NSCLC patients (*n* = 30) and indicated that CTCs in pulmonary veins predicted worsening survival [100]. Similarly, CTCs with pulmonary vein dissemination before tumour resection in NSCLC were associated with lung-cancer-specific recurrence and poor DFS [101]—An RCT compared vein-first ligation versus artery-first ligation on CTCs and survival outcomes in resected NSCLC patients. The ligation of pulmonary veins first prevented CTCs from entering the circulation, and patients in the vein-first ligation had significantly better DFS, OS, and lung-cancer-specific survival (LCSS) [163]. In conclusion, a meta-analysis by Wankhede et al. revealed that baseline and postoperative CTCs correlated with worsening DFS and OS in early-stage NSCLC [102]. Therefore, CTCs in preoperative and postoperative peripheral blood and pulmonary veins may result in tumour recurrence and metastasis, ultimately leading to a poor prognosis.

#### 3.3.2. Routine Examinations in Blood

Routine blood examinations have been performed on cancer patients during hospitalisation, and the results may reveal prognostic information about the patients.

##### Hemoglobin (Hb)

Regarding stage IB–II NSCLC patients, worsening OS was seen when baseline Hb was <120 g/L. By contrast, it had a tendency for favourable RFS in patients receiving adjuvant chemotherapy (ACT) with a nadir during-treatment Hb (<100 g/L) and favourable RFS and OS in patients with a maximum during-treatment decrease in Hb (>30%) [103].

##### Platelet (PLT)

Elevated PLT (>253 × 10^9^/L) and lymphocytes (>1.8 × 10^9^/L) in preoperative peripheral blood were independent prognostic factors for postoperative recurrence in pT1 NSCLC [104]. Hou et al. also revealed that high preoperative PLT (>215 × 10^9^/L) correlated with unfavourable PFS in early-stage NSCLC [105].

##### White Blood Cells

Several studies have examined the role of circulatory inflammatory markers in the prognosis of NSCLC patients. Elevated white blood cells above the median (8.6 × 10^9^ cells/L) were correlated with worsening RFS and OS in resected NSCLC (Stage I–IIIA) [106]. Kobayashi et al. confirmed that preoperative lymphocytes (cut-off values: 1900 mm^−3^) were independent favourable predictors for OS in resected NSCLC, but neutrophils (cut-off values: 4500 mm^−3^) were not [107]. A study involving 142 NSCLC patients (Stage IB–IIIA) concluded that preoperative lymphocytes (cut-off values: 1800 mm^−3^) were associated with a better DFS but not with OS, and neutrophils did not affect prognosis [108]. Conversely, Mitchell et al. concluded that preoperative elevated circulating neutrophils (>103 cells/µL) correlated with poor OS, but preoperative lymphocytes were not [109]. Another study examined the effect of giant peripheral tumour-macrophage fusion cells (TMFs) on the survival of stage I–IIIA NSCLC. More than one TMF and gigantic TMF sizes higher than or equal to 50 um were associated with worsening DFS and OS [110].

The circulating preoperative neutrophil–lymphocyte ratio (NLR) as a significant prognostic indicator was associated with reduced survival in early-stage NSCLC [111,112]. Another study confirmed that preoperative NLR (cut-off value: 3.3) was a marker for poor DFS and OS, whereas the platelet–lymphocyte ratio (PLR, cut-off values: 171) did not correlate with prognosis. Therefore, NLR was superior to PLR in terms of prognosis for pN0 NSCLC [113]. Seitlinger et al. focused on the preoperative and postoperative NLR in the prognosis of resected NSCLC. Both high NLR levels (>4.07) were associated with reduced OS and TTR [114]. A meta-analysis including 21 studies on the survival of early-stage NSCLC indicated that pretreatment-raised NLR (≥2.5) and PLR (≥150) were correlated with inferior DFS and OS [115]. Regarding the prognostic implications of the lymphocyte–monocyte ratio (LMR), a high baseline LMR (>3.68) may predict favourable DFS and OS in early-stage NSCLC [116]. The preoperative circulating uric acid-to-lymphocyte ratio (ULR, cut-off values: 3.83) was an independent predictor correlated with shortened DFS and OS in NSCLC patients (Stage I–II) [117]. Yan et al. combined neutrophil × platelet/lymphocyte to redefine the systemic immune-inflammation index (SII). High levels of preoperative SII (≥402.37) in resected NSCLC were indicative of worse DFS and OS [118].

#### 3.3.3. Circulating Nucleic Acids

Circulating nucleic acids, including several DNA or RNA molecules, originate from cell apoptosis and necrosis in the circulation [164]. Several studies confirmed that circulating DNA and RNA were prognostic predictors in resected NSCLC.

##### Circulating Tumour DNA (ctDNA)

CtDNA is cell-derived free DNA from tumour cells in the blood. Regarding the prognostic significance of perioperative ctDNA in early-stage NSCLC, the study (DYNAMIC) revealed that detectable ctDNA at three days and one month after surgery predicted worsening RFS and OS when compared to undetectable ctDNA [119]. Similarly, postoperatively detectable plasma ctDNA was a worse predictor for PFS and OS [120]. Wang et al. also demonstrated that detectable ctDNA was positively correlated with an increasing possibility of tumour recurrence or metastasis in 82 patients with stage I LUAD [121]. Another study by Qiu et al. classified stage II–III NSCLC patients by considering whether they received adjuvant chemotherapy (ACT) or had postoperative detectable ctDNA, and patients with high levels of ctDNA had significantly reduced RFS when compared to patients with low levels of ctDNA, whether they received ACT or not [122]. LUNGCA-1 confirmed that high levels of ctDNA preoperatively predicted poor RFS. The authors performed postoperative ctDNA-based molecular residual disease (MRD) detection and concluded that MRD indicated tumour recurrence [123]. Furthermore, several studies revealed that stage I–III NSCLC patients with both presurgical and postsurgical positive ctDNA had poorer survival than patients with negative ctDNA [124,125]. Therefore, perioperatively detectable ctDNA can predict reduced survival in patients with early-stage NSCLC.

##### MicroRNAs (MiRNAs)

Circulating miRNAs as noninvasive indicators still have prognostic value in early-stage NSCLC. Hu et al. developed four serum miRNAs (miR-486, miR-30d, miR-1, and miR-499) as a microRNA signature, revealing that high levels of miR-486 and miR-30d and low levels of miR-1 and miR-499 significantly correlated with poor OS in stage I–IIIA NSCLC [126]. Heegaard et al. studied the prognosis of paired plasma and serum miRNAs in stage I–II NSCLC. It concluded that only high plasma expression of miR-let-7b was associated with increased CSS, and all serum miRNAs did not correlate with patient prognosis [127]. Another study demonstrated that NSCLC patients (Stage I–III) with higher exosome miR-451a in plasma had reduced DFS and OS compared to patients with lower exosome miR-451a [128]. Han et al. isolated several miRNAs in extracellular vesicles (EV) from pulmonary tumour drainage veins (TDVs) and identified miR-203a-3p as a recurrence marker in the 18-patient screening cohort. They further verified that miR-203a-3p was a prognostic indicator for unfavourable TTR in the 70-patient validation cohort [129].

#### 3.3.4. Tumour Markers

The prognostic effect of tumour markers in early-stage NSCLC has been intensely reported.

##### Carcinoembryonic Antigen (CEA)

Sawabata et al. reported that pathological I–II NSCLC patients with high preoperative serum CEA (>7.0 ng/mL) exhibited significantly reduced five-year survival rates when compared to patients with normal CEA (five-year survival: high 49% vs. average 72%). Furthermore, patients with persistently high CEA after surgery had worse survival than those who returned to normal levels (five-year survival: high 18% vs. average 68%) [130]. Several studies also concluded that preoperative serum CEA significantly correlated with poor survival in resected NSCLC [131,132,133,134,135,136,137]. Hashinokuchi et al. even studied the combination of preoperative maximum standardised uptake value (SUVmax, cut-off values: 2.96) and CEA (cut-off values: 5.3 ng/mL) on the prognosis of stage IA LUAD. The survival time of patients with higher SC (SUVmax and CEA) levels was significantly reduced compared to patients with lower levels [138]. Regarding the predictive effects of postoperative CEA, serum elevated CEA was the significant determinant for worsening prognosis in stage I NSCLC [139,140,141,142]. In summary, high perioperative levels of CEA act as a predictor for poor survival in NSCLC patients.

##### CYFRA 21-1

Several studies have concluded that preoperative serum CYFRA21-1 was an independent predictor for unfavourable survival in early-stage NSCLC [135,143,144]. However, Kozu et al. revealed that preoperative CYFRA21-1 (cut-off values: 3.5 ng/mL) did not predict recurrence or survival in stage I NSCLC [136]. Mizuguchi et al. concluded that preoperative circulating CYFRA21-1 (cut-off values: 3.2 ng/mL) and Sialyl Lewisx (SLX, cut-off values: 36 U/mL) were independent predictors for poor survival in stage I NSCLC. Moreover, patients with the combination of CYFRA21-1 and SLX positivity had a significantly worse prognosis than those with negativity (five-year survival: positivity 13% vs. negativity 80%) [145]. The study by Muley et al. combined preoperative CYFRA21-1 (cut-off values: 3.3 ng/mL) and CEA (cut-off values: 9.8 ng/mL) to form a new parameter named tumour marker index (TMI) and demonstrated that elevated TMI (>0.54) negatively affected OS in stage I–II NSCLC patients [146].

##### Other Tumour Markers

Preoperative serum neuron-specific enolase (NSE), squamous cell carcinoma antigen (SCC-Ag), CA125, and CA199 had no significant effect on stage I NSCLC patients’ prognoses [147]. Reinmuth et al. also confirmed that NSE and SCC-Ag did not affect the survival of early-stage NSCLC (Stage I–IIIA) [143]. In contrast, a retrospective study demonstrated that preoperatively elevated NSE (>12.5 ng/mL) and CA125 (>35 U/mL) significantly correlated with poor DFS and OS in resected NSCLC patients. Furthermore, SCC-Ag (cut-off values: 1.5 ng/mL) did not affect all patients’ survival. However, SCC-Ag was only associated with reduced DFS and OS in LUSC patients [148]. Another retrospective study revealed that preoperative increased CA125 (>3.2 ng/mL) or CA199 (>3.2 ng/mL) correlated with poor RFS in resected LUAD patients (Stage IA-IIIA) [149]. Therefore, the prognostic value of circulating NSE, SCC-Ag, CA125, and CA199 in early-stage NSCLC remains debatable.

#### 3.3.5. Other Blood Biomarkers

##### C-Reactive Protein (CRP)

CRP is a marker for the circulating inflammatory response, and stage I–III patients with elevated preoperative CRP (>10 mg/L) had shorter CSS compared to patients with normal CRP (median survival: elevated 26.2 months vs. normal 75.9 months) [150]. Similarly, the preoperative levels of CRP ≥20 mg/L also correlated with worsening OS in stage I–II NSCLC, which was also confirmed in the validation group [151].

##### Fibrinogen and D-Dimer

Preoperative increased serum fibrinogen (≥4 g/L) was an effective predictor for worsening PFS and OS in resected NSCLC [152]. D-dimers are predictors of circulating fibrinolytic system activation. Assessing 232 patients with stage I–IIIA NSCLC, it was found that preoperative plasma D-dimer (cut-off values: 0.3 ug/mL) in plasma was an independent marker for reduced OS (one-year OS: high 76.5% vs. average 93.9%) [153]. Jiang et al. confirmed that preoperative and postoperative plasma high levels of fibrinogen (≥4 g/L) and D-dimer (≥0.55 mg/L) correlated with reduced DFS and OS in early-stage NSCLC [154]. However, Hou et al. confirmed that preoperative fibrinogen and D-dimer did not correlate with patient survival. This may be attributed to the short follow-up of 13.2 months [105].

##### Albumin

Kawai et al. revealed that preoperative (>23 mg/dL) and postoperative (>15 mg/dL) elevated albumin was an independent determinant for unfavourable DFS in resected NSCLC (Stage IA-IIIB) [155]. Preoperative circulating albumin and lymphocytes were combined as the prognostic nutritional index (PNI), and resected NSCLC patients with higher levels of PNI (≥48) had a greater prolonged RFS and OS than patients with lower levels [156]. Yamauchi et al. calculated the CRP–albumin ratio (CAR) to elucidate its prognostic effect in IIIA LUAD patients (*n* = 156), concluding that CAR (cut-off values: 0.6) was an unfavourable biomarker for RFS [157]. The study by Li et al. also confirmed that resected NSCLC patients with a preoperative albumin–globulin score (AGS) of 2 had significantly reduced DFS and OS than patients with an AGS of 0 or 1 [158]. For early-stage NSCLC patients receiving lobectomy, the preoperative albumin–globulin ratio (AGR, cut-off values: 1.76) independently predicted favourable survival [159].

##### Osteopontin (OPN)

Several investigators have examined the predictive effects of OPN in NSCLC patients receiving radical surgery. Patients with high preoperative serum levels of OPN (>81.3 ng/mL) had a poorer prognosis than patients with low levels (five-year survival: high 63.7% vs. low 82.0%) [160]. However, another study indicated that serum OPN did not affect the prognosis of resected NSCLC patients [161].

## 4. Conclusions

Treatment modalities for early-stage NSCLC have changed dramatically over the last decades, yet the postoperative survival of these patients remains unsatisfactory. Hence, we included mutant genes, immune-related indicators from resected tumours, and blood biomarkers to explore their prognostic effect on NSCLC. The prognostic impact of alterations on the EGFR, ALK, MET, ROS1, or TP53 in resected NSCLC remains debatable for mutant genes. KRAS alterations indicate unfavourable outcomes in early-stage NSCLC. Regarding early-stage NSCLC with EGFR alterations, adjuvant or neoadjuvant EGFR-TKIs can significantly improve patient prognosis. Due to limited data on other driver genes with low alteration frequencies, the forecast can only be assessed with further studies. Based on current studies, resected NSCLC patients with high levels of PD-L1 exhibited lower survival than those with low levels. Conversely, PD-L1 or PD-1 inhibitors can substantially increase patient survival. Other immune checkpoints (PD-1, PD-L2, and TIM-3) also lacked prognostic value owing to the limited number of studies. For circulating indicators, preoperative and postoperative peripheral venous CTCs and pulmonary venous CTCs predicted unfavourable prognoses, leading to distant metastases in NSCLC. Similarly, patients with detectable perioperative ctDNA also had inferior survival. CEA, as a circulating tumour marker, has been studied extensively. Patients with preoperatively and postoperatively elevated CEA in the circulation predicted significantly reduced survival outcomes.

Several genetic alterations and immune and circulatory indicators show superior predictions for the prognoses and treatment responses of early-stage NSCLC. Autonomous artificial intelligence (AI) systems combined with CT images can provide noninvasive whole-lung analyses to forecast the projections of lung cancer patients receiving EGFR-TKI therapy [165]. Additionally, the AI algorithms identified 32 preoperative markers in the blood, including CA125, CA199, and CRP, to predict the outcomes of epithelial ovarian cancer [166]. In the future, we will incorporate mutated genes, immune checkpoints, and blood-based biomarkers by applying AI to construct new prognostic models that predict patient outcomes accurately and guide individualised treatment (Figure 1). Furthermore, combined models as biological factors should be incorporated into the tumour–node-metastasis (TNM) staging system.

## Figures and Tables

**Figure 1 cancers-15-04561-f001:**
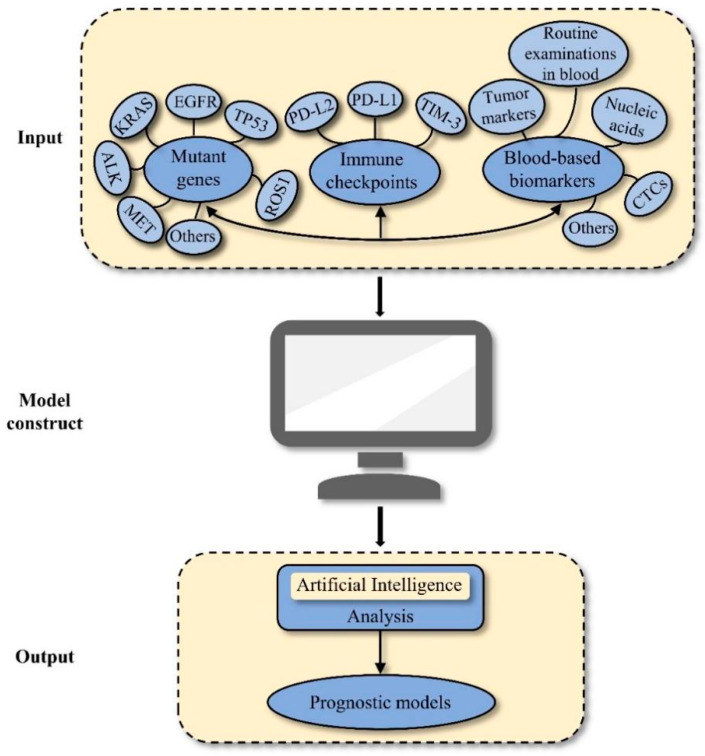
Flowchart of constructing prognostic models by applying AI. AI, artificial intelligence.

**Table 1 cancers-15-04561-t001:** Summary of studies relating to genetic alterations in resected early-stage NSCLC.

Mutant Genes	Ref. (NOP)	Study Design	Histology	Phase	Ethnicity	Median Follow-Up (Months)	Percentage of Genetic Alterations	Prognostic Implication
**EGFR**								
EGFR alterations	[9] (374)	RCT	Ns-NSCLC	II–IIIA	Japanese	NA	37.2%	Worse RFS
EGFR alterations	[10] (1512)	R	LUAD	I–III	Chinese	53.2	61.8%	Worse RFS
EGFR alterations	[11] (1155)	R	LUAD	I–IIIA (N0-1)	Japanese	36	50.6%	Worse RFS
EGFR amplification	[12] (637)	P	LUAD	I–IIIA (≤3 cm)	Chinese	30.57	5.8%	Worse RFS
EGFR amplification	[13] (109)	R	NSCLC	I–III	Japanese	NA	5.5%	Worse OS
EGFR alterations	[14] (237)	P	LUAD	I–III	Japanese	NA	45.5%	Improved RFS
EGFR alterations	[15] (307)	R	NSCLC	I	American	30	20.2%	Improved DFS and OS
EGFR alterations	[16] (164)	R	NSCLC	I–IIIA	Chinese	NA	31.7%	Improved OS
EGFR alterations	[17] (1231)	R	NSCLC	I	Japanese	36	44.3%	IA: No impact on survival; IB: Improved DSS and OS
EGFR exons 18-19	[18] (133)	P	NSCLC	I–III	Korean	29	7.5%	Improved OS
EGFR exon 19	[19] (276)	R	LUAD	I–III	French/Italian	46	55.1%	Stage I: No impact on OS; Stage II/III: Improved OS
EGFR exon 19	[20] (202)	R	LUAD	II–IIIA (N1-2)	Japanese	NA	27.2%	Improved DFS and OS
EGFR exon 19	[21] (835)	R	LUAD	I–III	Japanese	47	21%	Worse RFS
EGFR T790M	[22] (373)	P	NSCLC	I–IIIB	Japanese	48.6	79.9%	Worse RFS and OS
Phosphorylated EGFR	[23] (49)	P	NSCLC	I	Austrian	29.3	NA	Worse survival
**KRAS**								
KRAS G12C	[24] (604)	P	LUAD	I–III	American	30.1	16%	Worse DFS
KRAS alterations	[25] (1118)	R	LUAD	I–III	American	27	25%	No impact on OS
KRAS alterations	[26] (312)	R	LUAD	I	American	36	41%	Worse DFS and OS
KRAS alterations	[27] (482)	R	LUAD	I–II	American	56.8	27%	Worse OS
KRAS alterations	[28] (324)	R	LUAD	I–III	American	33.67	38.1%	Worse DFS and OS
KRAS alterations	[16] (164)	R	NSCLC	I–IIIA	Chinese	NA	4.3%	Worse OS
KRAS alterations	[29] (216)	R	NSCLC	I–IIIA	Spanish	14.25	13.4%	Improved DSS
**ALK**								
ALK rearrangement	[30] (124)	R	LUAD	I	Japanese	NA	19.4%	Worse RFS; No impact on OS
ALK rearrangement	[31] (764)	R	NSCLC	I–III	American	60.5	4%	Worse RFS; No impact on OS
ALK rearrangement	[32] (1925)	R	LUAD	I–IIIB	Japanese	72	3.9%	Improved CSS and OS
EML4-ALK fusion variant 3	[33] (55)	R	NSCLC	I–III	Chinese	34.8	34.6%	Worse DFS
ALK rearrangement	[34] (231)	R	LUAD	I–IIIB	Korean	26	9%	No impact on DFS and OS
ALK rearrangement	[35] (735)	R	NSCLC	I–III	Korean	45	3.8%	No impact on DFS and OS
**MET**								
MET alterations	[36] (311)	R	LUAD	IB–IIIA	Korean	22.8	45.3%	Worse RFS and OS in EGFRmut (-) group
MET alterations	[37] (265)	R	NSCLC	I–II	Italian	40	NA	Worse OS
MET amplification	[38] (1540)	R	NSCLC	I–III	International	57.3	22.8%	No impact on OS
METex14 skipping	[39] (795)	R	LUAD	I–III	Korean	NA	2.1%	No impact on OS
**ROS1**								
ROS1 fusion	[40] (230)	R	LUAD	I–II	Korean	49	NA	Worse RFS
ROS1 fusion	[41] (160)	P	LUAD	I–IIIA	Chinese	50.1	3.1%	No impact on OS
**TP53**								
TP53 alterations	[12] (637)	P	LUAD	I–IIIA (≤3 cm)	Chinese	30.57	28.1%	Worse RFS
TP53 alterations	[40] (230)	R	LUAD	I–II	Korean	49	18.3%	Worse RFS
TP53 alterations	[42] (214)	R	NSCLC	I–III	Canadian	54	50%	Worse DFS and OS
TP53 alterations	[43] (50)	R	NSCLC	IA	American	NA	NA	LUAD: Worse OS; No impact on DFS; LUSC: No effect on DFS and OS
p53 pathway alterations	[44] (492)	P	LUAD	I–III	American	19	NA	Worse DFS
p53 alterations	[45] (76)	P	NSCLC	I–IIIB	Italian	23	54%	Worse DFS
p53 alterations	[46] (102)	P	NSCLC	I–IIIA	Polish	28	NA	Worse OS
**Others**								
RET fusion	[40] (230)	R	LUAD	I–II	Korean	49	NA	Worse RFS
BRAF alterations	[28] (324)	R	LUAD	I–III	American	33.67	5.9%	Worse DFS and OS
PIK3CA alterations	[47] (308)	P	LUSC	I–III	Norwegian	67.6	11.4%	Improved OS and TTR
PIK3CA alterations	[48] (810)	R	LUAD	I–IIIA	Chinese	NA	2.8%	Worse OS
PIK3CA alterations	[49] (167)	R	NSCLC	I	Italian	120	3.6%	Worse survival
TTF-1 amplification	[50] (118)	R	LUAD	I–III	Korean	NA	12.7%	Worse DFS and OS
PKD and TP53 co-alterations	[51] (147)	P	LUAD	I–III	Chinese	31.3	7%	Worse DFS; No impact on OS

Abbreviations: EGFR, epidermal growth factor receptor; KRAS, Kirsten rat sarcoma viral oncogene homologue; ALK, anaplastic lymphoma kinase; MET, mesenchymal-epithelial transition; METex14, MET exon 14; ROS1, C-ros oncogene 1; TP53, tumour protein p53; RET, rearranged during transfection; BRAF, V-RAF murine sarcoma viral oncogene homolog B1; TTF-1, thyroid transcription factor-1; PKD, protein kinase D; Ref, reference; NOP, number of the population; RCT, randomised controlled trial; R, retrospective study; P, prospective study; NSCLC, non-small-cell lung cancer; Ns-NSCLC, nonsquamous non-small-cell lung cancer; LUAD, lung adenocarcinoma; LUSC, lung squamous cell carcinoma; NA, not available; DFS, disease-free survival; RFS, relapse-free survival; OS, overall survival; DSS, disease-specific survival; R, retrospective study; P, prospective study; Meta, meta-analysis; TTR, time to recurrence.

**Table 2 cancers-15-04561-t002:** Summary of studies relating to immune-related indicators in resected early-stage NSCLC.

Immune-Related Indicators	Ref. (NOP)	Study Design	Histology	Phase	Ethnicity	Median Follow-Up (Months)	Percentage of Indicators Overexpression	Prognostic Implication
PD-L1	[70] (316)	R	LUAD	I–III	Korean	NA	18.6%	Worse RFS and OS
PD-L1	[71] (193)	R	LUAD	I–III	Korean	NA	21.2%	Worse RFS and OS
PD-L1; PD-L2	[72] (433)	R	LUAD	I–III	Japanese	NA	NA	Worse DFS and OS
PD-L1	[73] (417)	R	LUAD	I–III	Japanese	NA	34.5%	Worse DFS and OS
PD-L1	[74] (205)	R	LUSC	I–III	Japanese	NA	51.7%	Worse OS
PD-L1	[75] (164)	R	NSCLC	I–IIIB	Japanese	55.6	50%	Worse OS
PD-L1	[76] (455)	R	NSCLC	I–IIIA	Asian	47	NA	Worse DFS and OS
PD-L1	[77] (136)	R	NSCLC	I–III	Korean	84	25.7%	Worse RFS and OS
PD-L1	[78] (126)	R	NSCLC	I	Japanese	47.5	18.3%	CD8-high cohort: No impact on RFS; CD8-low cohort: Worse RFS
PD-L1	[79] (163)	R	LUAD	I	Chinese	71	39.9%	Improved RFS; No impact on OS
PD-L1	[80] (678)	R	NSCLC	I–III	Australian	NA	7.4%	Improved OS
PD-L1	[81] (228)	R	NSCLC	I–IIIA	Japanese	NA	24.1%	Stage I: Improved RFS; Stage II–IIIA: Worse RFS
PD-L1	[82] (386)	R	LUAD	I–III	Chinese	54	48.3%	No impact on DFS and OS
PD-L1	[83] (3790)	Meta	NSCLC	I–III	International	NA	NA	Worse DFS and OS
TIM-3	[84] (223)	R	LUAD	I–III	Chinese	NA	48%	Worse RFS and OS
TMB	[85] (92)	R	NSCLC	I–III	Canadian	NA	NA	Worse OS
TMB	[86] (78)	R	LUAD	I–II	Caucasian	56.1	NA	No impact on RFS and OS
OX40	[87] (139)	R	NSCLC	I	Polish	NA	NA	Worse RFS and OS
OX40	[88] (100)	R	NSCLC	I–III	American	NA	NA	Improved OS

Abbreviations: PD-L1, programmed death-ligand 1; PD-L2, programmed death-ligand 2; TIM-3, T-cell immunoglobulin and mucin-domain-containing-3; TMB, tumour mutation burden; Ref, reference; NOP, number of the population; R, retrospective study; P, prospective study; Meta, meta-analysis; NSCLC, non-small-cell lung cancer; LUAD, lung adenocarcinoma; LUSC, lung squamous cell carcinoma; NA, not available; DFS, disease-free survival; RFS, relapse-free survival; OS, overall survival.

**Table 3 cancers-15-04561-t003:** Summary of studies relating to blood-based biomarkers in resected early-stage NSCLC.

Blood-Based Biomarkers	Ref. (NOP)	Study Design	Histology	Phase	Ethnicity	Median Follow-Up (Months)	Prognostic Implications
**CTCs**							
CTCs	[93] (40)	P	LUAD	I–IIIA	Austrian	16	Worse DFS
CTCs	[94] (137)	P	NSCLC	I–II	French	NA	Worse DFS and OS
FR^+^ CTCs	[95] (52)	P	NSCLC	I–III	Chinese	NA	Worse RFS and OS
PD-L1^+^/EMT^+^ CTCs	[96] (30)	P	NSCLC	I–IIIA	American	14.3	Worse RFS and OS
CTCs	[97] (97)	P	NSCLC	I–III	Spanish	LAUD: 30.5; LUSC: 32	LUAD: Worse RFS and OS; LUSC: No impact on survival
TTF-1^+^ CTCs	[98] (79)	P	NSCLC	I–III	Korean	NA	Worse PFS
LUNX^+^ CTCS	[99] (68)	P	NSCLC	I–IIIA	Chinese	39.5	Worse DFS and OS
CTCs	[100] (30)	P	NSCLC	I–III	British	22	Worse DFS and OS
CTCs	[101] (100)	P	NSCLC	I–III	British	33.1	Worse DFS
CTCs	[102] (1321)	Meta	NSCLC	I–IIIA	International	Range: 13–84	Worse DFS and OS
**Routine examinations of blood**							
Hb	[103] (482)	R	NSCLC	IB–II	Canadian	NA	Baseline Hb: Worse OS; During-treatment Hb: Improved RFS and OS
PLT; Lymphocyte	[104] (103)	R	NSCLC	I	American	42	Worse RFS
PLT	[105] (395)	P	NSCLC	I–III	Chinese	13.2	Worse PFS
White blood cells	[106] (335)	P	NSCLC	I–IIIA	Danish	NA	Worse RFS and OS
Lymphocyte; Neutrophil	[107] (237)	R	NSCLC	I–III	Japanese	NA	Lymphocyte: Improved OS; Neutrophil: No impact on OS
Lymphocyte; Neutrophil	[108] (142)	R	NSCLC	IB–IIIA	Chinese	NA	Lymphocyte: Improved DFS/No impact on OS; Neutrophil: No impact on survival
Neutrophil; Lymphocyte	[109] (1524)	P	NSCLC	I–IIIA	American	60.7	Neutrophil: Worse OS; Lymphocyte: No impact on OS
TMFs	[110] (115)	P	NSCLC	I–IIIA	Colombian	26.6	Worse DFS and OS
NLR	[111] (343)	R	NSCLC	I	Japanese	73.5	Worse RFS and OS
NLR	[112] (952)	R	NSCLC	I–III	Chinese	40	Worse OS
NLR; PLR	[113] (400)	R	NSCLC	I–II (N0)	Chinese	46	NLR: Worse DFS and OS; PLR: No impact on survival
NLR	[114] (2027)	R	NSCLC	I–III	French	69	Worse OS and TTR
NLR; PLR	[115] (14,242)	Meta	NSCLC	I–III	International	NA	Worse DFS and OS
LMR	[116] (1453)	R	NSCLC	I–III	Chinese	NA	Improved DFS and OS
ULR	[117] (335)	P	NSCLC	I–II	Chinese	51	Worse DFS and OS
SII	[118] (538)	R	NSCLC	I–IIIA	Chinese	54	Worse DFS and OS
**Circulating nucleic acids**							
ctDNA	[119] (205)	P	NSCLC	I–IIIA	Chinese	17.6	Worse RFS and OS
ctDNA	[120] (33)	P	NSCLC	I–IIIB	German	26.2	Worse PFS and OS
ctDNA	[121] (82)	P	LUAD	I	Chinese	22.83	Worse DFS
ctDNA	[122] (89)	P	LUAD	IIB–IIIA	Chinese	NA	Worse RFS
ctDNA	[123] (330)	P	NSCLC	I–III	Chinese	35.6	Worse RFS
ctDNA	[124] (119)	P	NSCLC	I–IIIA	Chinese	30.7	Worse RFS and OS
ctDNA	[125] (88)	P	NSCLC	I–III	British	36	Worse RFS and OS
Four-miRNA signature (miR-486 and miR-30d, miR-1, miR-499)	[126] (303)	R	NSCLC	I–IIIA	Chinese	NA	High miR-486 and miR-30d; Low miR-1 and miR-499: Worse OS
MiR-let-7b	[127] (220)	R	NSCLC	I–II	American	NA	Improved CSS
MiR-451a	[128] (285)	R	NSCLC	I–III	Japanese	24	Worse DFS and OS
MiR-203a-3p	[129] (88)	R	NSCLC	I–III	Spanish	Screening/Validation cohort: 66.07/54.23	Worse TTR
**Tumour markers**							
CEA	[130] (253)	R	NSCLC	I–II	Japanese	NA	Worse OS
CEA	[131] (265)	R	NSCLC	I–IIIB	Japanese	43	Worse DSS and OS
CEA	[132] (65)	R	LUAD	IA	Japanese	NA	Worse DFS and OS
CEA	[133] (45)	R	NSCLC	T1-2N1M0	Japanese	39.8	Worse OS
CEA	[134] (87)	R	LUAD	I–IIIB (≤3 cm)	Japanese	NA	Worse DFS
CEA; CYFRA 21-1	[135] (341)	R	NSCLC	I	Japanese	NA	Worse OS
CEA; CYFRA 21-1	[136] (467)	R	NSCLC	I	Japanese	54	CEA: Worse DFS and OS; CYFRA 21-1: No impact on survival
CEA	[137] (758)	R	NSCLC	I	Chinese	NA	Worse PFS
Combination of SUVmax and CEA	[138] (410)	R	LUAD	IA	Japanese	NA	Worse RFS and OS
CEA	[139] (242)	R	NSCLC	IA	Japanese	64	Worse survival
CEA	[140] (250)	R	LUAD	I	Japanese	73.2	Worse DFS
CEA	[141] (600)	R	LUAD	IB	Chinese	46.7	Worse RFS and OS
CEA	[142] (136)	R	LUAD	I	Japanese	28.3	Worse RFS
CYFRA 21-1; NSE; SCC-Ag	[143] (67)	P	NSCLC	I–IIIA	German	86	CYFRA 21-1: Worse DFS and OS; NSE; SCC-Ag: No impact on survival
CYFRA 21-1	[144] (298)	R	LUAD	I–III	Korean	43.3	Worse DFS and OS
Combination of CYFRA 21-1 and SLX	[145] (137)	R	NSCLC	I	Japanese	NA	Worse survival
TMI	[146] (261)	R	NSCLC	I–II	German	59.8	Worse OS
NSE; SCC-Ag; CA125; CA199	[147] (164)	R	NSCLC	I	Chinese	26	No impact on survival
NSE; CA125; SCC-Ag	[148] (481)	R	NSCLC	I–IIIB	Chinese	NA	NSE; CA125: Worse DFS and OS; SCC-Ag: No impact on survival
CA125; CA199	[149] (58)	R	LUAD	I–IIIA	Swedish	39.6	Worse RFS
**Other blood biomarkers**							
CRP	[150] (96)	R	NSCLC	I–III	British	75	Worse CSS
CRP	[151] (229)	R	NSCLC	I–II	French	89.2	Worse OS
Fibrinogen	[152] (567)	R	NSCLC	I–IIIB	Chinese	NA	Worse PFS and OS
D-dimer	[153] (232)	R	NSCLC	I–IIIA	Chinese	47	Worse OS
Fibrinogen; D-dimer	[154] (184)	R	NSCLC	I–IIIA	Chinese	18.5	Worse DFS and OS
Fibrinogen; D-dimer	[105] (395)	P	NSCLC	I–III	Chinese	13.2	Worse PFS
Prealbumin	[155] (44)	P	NSCLC	I–III	Japanese	NA	Improved DFS
PNI	[156] (248)	R	NSCLC	I–III	Japanese	51	Improved RFS and OS
CAR	[157] (156)	R	NSCLC	IIIA (N2)	German	NA	Worse RFS
AGS	[158] (312)	R	NSCLC	I–III	Chinese	41	Worse DFS and OS
AGR	[159] (180)	R	NSCLC	I–III	Chinese	NA	Improved OS
OPN	[160] (244)	R	NSCLC	I–III	Japanese	NA	Worse survival
OPN	[161] (201)	R	NSCLC	I–III	Norwegian	34	No impact on survival

Abbreviations: CTCs, circulating tumour cells; Hb, hemoglobin; PLT, platelet; TMFs, tumour-macrophage fusion cells; NLR, neutrophil–lymphocyte ratio; PLR, platelet–lymphocyte ratio; LMR, lymphocyte–monocyte ratio; ULR, uric acid-to-lymphocyte ratio; SII, systemic immune-inflammation index (SII); ctDNA, circulating tumour DNA; CEA, carcinoembryonic antigen; SLX, Sialyl Lewisx; TMI, tumour marker index; NSE, neuron-specific enolase; SCC-Ag, squamous cell carcinoma antigen; CRP, C-reactive protein; PNI, prognostic nutritional index; CAR, CRP–albumin ratio; AGS, albumin–globulin score; AGR, albumin–globulin ratio; OPN, osteopontin; Ref, reference; NOP, number of the population; R, retrospective study; P, prospective study; Meta, meta-analysis; NSCLC, non-small-cell lung cancer; LUAD, lung adenocarcinoma; LUSC, lung squamous cell carcinoma; NA, not available; DFS, disease-free survival; RFS, relapse-free survival; PFS, progression-free survival; OS, overall survival; TTR, time to recurrence; DSS, disease-specific survival; CSS, cancer-specific survival.

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
