# Peer review of "A Review of Biomarkers and Their Clinical Impact in Resected Early-Stage Non-Small-Cell Lung Cancer"

_cancers, 2023, doi:10.3390/cancers15184561_

Round 1
Reviewer 1 Report
This is a very thorough review of 167 publications that studied the effects of various pre- and peri- operative markers such as gene alterations and circulating factors on recurrence and survival in patients with NSCLC and LUAD.
The results of studies were stated, but little context provided. Most of the sections were statements taken from the papers, but this provides little in the way of information for readers, especially clinicians looking for a consensus. Most of the study participants were Asian and that can definitely be a factor when considering non-Asian populations. It would have been helpful to have a summary statement for many of the sections, especially with regard to how targeted therapies would affect the results. For example, is c-MET expression important or is activation of cMET by phosphorylation be more predictive. What about agents that target specific mutations, for example, EGFR? Does having a targeted agent result in better survival compared to someone who has the wild type gene, but no treatment options for relapse? Are any of these findings applicable to any other cancers? Are any of these potentially prognostic markers targetable? It is not always helpful to patients if they are told they have a high risk mutation without a treatment option. When discussing CTCs. it should be said that these CAN, not WILL result in recurrence, since it may take 5 million circulating breast cancer cells to form 1 metastatic lesion. It is not clear what is meant by "peri-operative PLT". For many markers, it may be a yes or no result, but what platelet levels are designated as high or low and which is important? This same is true for the discussion of other blood cell populations. How is tumor-specific circulating DNA identified? Some treatments result in an increase of circulating cell free DNA, but this can come from any cell that is injured or killed, beyond tumor cells. And, in fact, many tumors do not have known, signature mutations or the frequency of a specific mutation may be low.
The text needs to be expanded to improve clarity of many of these statements.
In general, the English is pretty good, but will need some minor editing. For example, the use of Spaniard and Pollack as ethnic designations would not be generally used. Certainly, Spanish or Polish are better terms, but maybe European or Caucasian is more indicative, or change the table column title to something else. That so many of the studies cited seem to have enrolled predominantly Asian participants may biased the results and should be addressed by the authors.
Using the word "achieved" in several places followed by "worse prognosis" or "reduced survival" makes the reader wonder at the intent. Native English speakers will expect that, to achieve something would be better than to not achieve. In addition, tissue that is stained for a marker is positive or negative. Negativity and positivity terms are not generally used. In fact, many expression assays are now categorized by high or low, not just positive or negative. This can have treatment consequences: in countries where patients rely on private insurance, like the US, some of this can determine if the treatment will be covered by the policy or not.
Reviewer 2 Report
Major comments:
1. Your future plan is not necessary addressed in this study such as in Simple Summary, Abstract, Conclusions, and Figure 1, particularly there is no AI results reviewed in this study. The concept is interesting if you can review some AI-based publications.
2. Can you summarize which biomarkers are suitable for what kind of therapies in patients with lung cancer.
Reviewer 3 Report
Cao et al. describing a review regarding biomarkers and their clinical impact in resected early-stage non-small cell lung cancer including mutations such as EGFR, ALK, MET, KRAS and PDL-1. I have few comments:
The article is well organized and very well written.
The article has a lot of information's and References from Asian articles and researches, I know that the EGFR for example is found more in Asian population but still most of the articles are worldwide, and as it is known that those diseases are founded all over the world.
Another point of view an article that changed the world for adjuvant EGFR- "ADAURA" trial was the first TKI as anti- EGFR that showed overall survival, and was published recently in 6/2023 as I remember in New England journal of medicine. I know that the deadline for the literature search was February 1st, 2023, but still important to add such a wonderful information's that changed the game of adjuvant setting in lung cancer.
I think more missing information's were published but the authors didn’t notice it, and could be added for the following subjects EGFR and MET, and Immunotherapy-
As an example the following article and reviews:
- Herbst RS, Wu YL, John T, Grohe C, Majem M, Wang J, Kato T, Goldman JW, Laktionov K, Kim SW, Yu CJ, Vu HV, Lu S, Lee KY, Mukhametshina G, Akewanlop C, de Marinis F, Bonanno L, Domine M, Shepherd FA, Urban D, Huang X, Bolanos A, Stachowiak M, Tsuboi M. Adjuvant Osimertinib for Resected EGFR-Mutated Stage IB-IIIA Non-Small-Cell Lung Cancer: Updated Results From the Phase III Randomized ADAURA Trial. J Clin Oncol. 2023 Apr 1;41(10):1830-1840. doi: 10.1200/JCO.22.02186. Epub 2023 Jan 31. Erratum in: J Clin Oncol. 2023 Aug 1;41(22):3877. PMID: 36720083; PMCID: PMC10082285.
- Tsuboi M, Herbst RS, John T, Kato T, Majem M, Grohé C, Wang J, Goldman JW, Lu S, Su WC, de Marinis F, Shepherd FA, Lee KH, Le NT, Dechaphunkul A, Kowalski D, Poole L, Bolanos A, Rukazenkov Y, Wu YL; ADAURA Investigators. Overall Survival with Osimertinib in Resected EGFR-Mutated NSCLC. N Engl J Med. 2023 Jul 13;389(2):137-147. doi: 10.1056/NEJMoa2304594. Epub 2023 Jun 4. PMID: 37272535.
- Szeto CH, Shalata W, Yakobson A, Agbarya A. Neoadjuvant and Adjuvant Immunotherapy in Early-Stage Non-Small-Cell Lung Cancer, Past, Present, and Future. J Clin Med. 2021 Nov 29;10(23):5614. doi: 10.3390/jcm10235614. PMID: 34884316; PMCID: PMC8658154.
- Shalata W, Yakobson A, Weissmann S, Oscar E, Iraqi M, Kian W, Peled N, Agbarya A. Crizotinib in MET Exon 14-Mutated or MET-Amplified in Advanced Disease Non-Small Cell Lung Cancer: A Retrospective, Single Institution Experience. Oncology. 2022;100(9):467-474. doi: 10.1159/000525188. Epub 2022 Jun 9. PMID: 35679833.
- Wu YL, John T, Grohe C, Majem M, Goldman JW, Kim SW, Kato T, Laktionov K, Vu HV, Wang Z, Lu S, Lee KY, Akewanlop C, Yu CJ, de Marinis F, Bonanno L, Domine M, Shepherd FA, Zeng L, Atasoy A, Herbst RS, Tsuboi M. Postoperative Chemotherapy Use and Outcomes From ADAURA: Osimertinib as Adjuvant Therapy for Resected EGFR-Mutated NSCLC. J Thorac Oncol. 2022 Mar;17(3):423-433. doi: 10.1016/j.jtho.2021.10.014. Epub 2021 Nov 2. PMID: 34740861.
Round 2
Reviewer 2 Report
No more questions.
Reviewer 3 Report
The review is well reorganized and more relevant and important informations were added.
The authors had made all of my suggestions. I have nothing to add.
good luck.